

# Temperature-mediated feeding between spring-associated and riverine-associated congeners, with implications for community segregation

Cody A. Craig, Jeremy D. Maikoetter and Timothy H. Bonner

Department of Biology/Aquatic Station, Texas State University, San Marcos, TX, USA

Corresponding author
Cody A. Craig, cac300@txstate.edu

## ABSTRACT

Freshwater fish communities segregate along water temperature gradients attributed in part to temperature-mediated physiological processes that affect species fitness. In spring complexes of southwest USA, spring complexes with narrow range of water temperatures are dominated by a community of fishes (i.e., spring-associated fishes), whereas riverine habitats with wide-range of water temperatures are dominated by a different community of fishes (i.e., riverine-associated fishes). The purpose of this study was to test a prediction of the concept that temperature-mediated species performance is a mechanism in maintaining community segregation. We predicted that a spring-associated fish (Largespring Gambusia *Gambusia geiseri*) would feed first and more often in a pairing with a riverine-associated fish (Western Mosquitofish *G. affinis*) at an average spring temperature (23 °C) and that the riverine-associated fish would feed first and more often in a pairing with the spring-associated fish at a warm riverine temperature (30 °C). Among four trails consisting of 30 pairings, at the spring complex temperature (23 °C), Largespring Gambusia had a greater number of first feeds (mean ± 1 SD, 5.0 ± 0.82) than Western Mosquitofish (2.5 ± 1.73) and had greater mean number of total feeds (1.9 ± 0.31) than Western Mosquitofish (0.81 ± 0.70). At the riverine environment temperature (30 °C), Western Mosquitofish had a greater number of first feeds (5.25 ± 1.71) than Largespring Gambusia (2.5 ± 1.73) and had greater mean number of total feeds (2.78 ± 1.05) than Largespring Gambusia (0.94 ± 0.68). Our findings suggest that temperature-mediated species performance could be maintaining segregation between the two fish communities. This study benefits our understanding of distributional patterns and improves threat assessments of stenothermal aquatic organisms.

## INTRODUCTION

Aquatic species and communities are distributed along altitudinal, geographical, and longitudinal gradients where habitats, food resources, predation, and water quality

conditions differ (*Vannote et al., 1980*; *Taniguchi & Nakano, 2000*). Among freshwater fishes, water temperature is one of several described mechanisms regulating distributional patterns (*Grossman & Freeman, 1987*). Temperature can influence interspecific interactions within freshwater fish communities when species temperature tolerances are overlapping (*Taniguchi et al., 1998*). Temperature-mediated interactions and its influence on species distributions, though difficult to quantify in nature (*Gerking, 1994*), are supported in laboratory experiments. *Taniguchi et al. (1998)* and *Taniguchi & Nakano (2000)* compared water temperature tolerances and behaviors among fishes distributed along an altitudinal gradient and found fish that inhabit cooler water at higher altitudes were more aggressive, consumed more food, had faster growth, and greater survival rate at cooler temperatures than lower altitude fishes. Conversely, fishes, which inhabit warmer water at lower altitudes, were more aggressive, consumed more food, and had faster growth at warmer temperatures than the higher altitude fish. *Carmona-Catot, Magellan & Garcia-Berthou (2013)* quantified pairwise feeding performance at three temperatures (i.e., 19, 24, and 29 °C) of an introduced warm-water cyprinodont and a native cool-water cyprinodont to assess temperature-mediated interactions in non-native species range expansion and native species extirpation potential. The invasive warm-water cyprinodont had a lower food capture rate compared to the native cool-water cyprinodont at the coolest temperature. At warmer temperatures, the invasive warm-water cyprinodont had a greater food capture rate compared to native cool-water cyprinodont. Laboratory results of temperature-mediated interactions suggest water temperature regulates fish distributions.

Spring complexes within limestone formations of southwest USA are evolutionary refugia with stenoecious water quality, including thermally-constant water temperatures (i.e., stenothermal habitat; range 21.0–23.3 °C), and distinct fish communities consisting of spring-associated fishes that have greater relative abundances and densities within spring complexes (*Craig et al., 2016*). As spring complexes transition downstream into riverine environments with less thermally-constant water temperatures (i.e., eurythermal habitat; range 6–30 °C) attributed to ambient conditions and merging with higher order streams, relative abundances and densities of spring-associated fishes are reduced and different species of fishes (i.e., riverine-associated fishes) become dominant. Similar to altitudinal gradients (*Taniguchi et al., 1998*; *Taniguchi & Nakano, 2000*), water temperature is a suggested mechanism in regulating richness, abundances, and densities of spring-associated fishes and riverine-associated fishes (*Hubbs, 1995*; *Kollaus & Bonner, 2012*) with spring-associated fishes being potentially more fit in stenothermal habitats and riverine-associated fishes being potentially more fit in eurythermal habitats. Dissimilar to altitudinal gradients, spring-associated fishes and riverine-associated fishes do not represent previously researched cold-water and warm-water forms with overlapping tolerances, but both are warm-water forms having similar temperature tolerances (*Hagen, 1964*; *Brandt et al., 1993*), and similar reproductive tolerances (*Bonner et al., 1998*; *McDonald et al., 2007*). In marine systems, species in stenothermal habitats might select away from eurythermal enzymes and proteins (*Graves & Somero, 1982*)

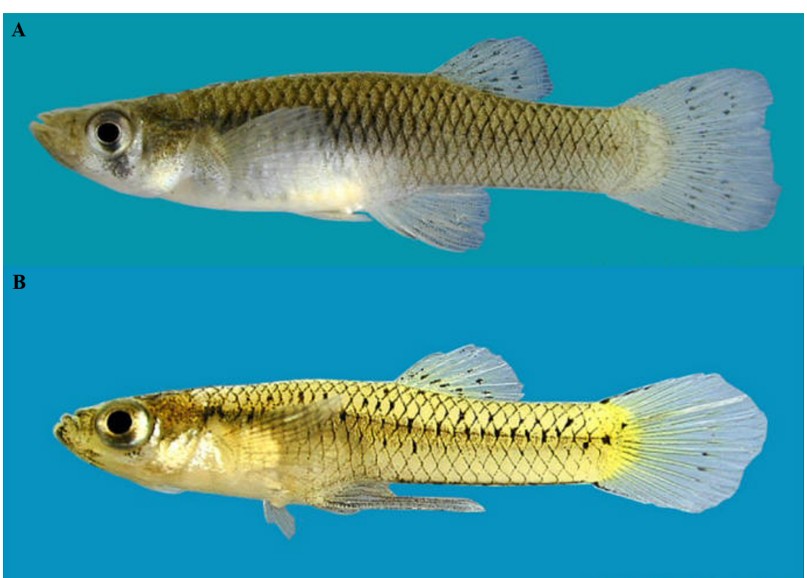

**Figure 1** *Gambusia affinis* **and** *Gambusia geiseri.* Example of species (A) *Gambusia affinis* female and (B) *Gambusia geiseri* male used in study. Picture from *Thomas, Bonner & Whiteside (2007)*.

and select for proteins and enzymes that are more energy efficient within a narrow range of temperatures (*Pörtner, Peck & Somero, 2007*), whereas species in eurythermal habitats are suggested to conserve temperature-dependent enzymes and proteins that enable tolerance of wide-ranging water temperatures (*Somero, Dahlhoff & Lin, 1996*). Differences in fitness between spring-associated fishes in stenothermal habitats and riverine-associated fishes in eurythermal habitats could explain patterns in fish community segregation in spring-river systems.

The purpose of this study was to test predictions of the concept that stenothermal habitat of spring complexes is a factor in maintaining community segregation between spring-associated and riverine-associated fishes. The study objective was to quantify feeding (i.e., first feed and number of total feeds) as a measure of performance between spring-associated and riverine-associated fish pairs at two water temperatures and determine if water temperatures favored one species over the other. Water temperatures selected were 23 °C, a typical water temperature in spring complexes, and 30 °C, a typical summertime temperature in riverine environments. We used two congeneric species, the riverine-associated Western Mosquitofish, *Gambusia affinis* (Fig. 1A) and the spring-associated Largespring Gambusia, *G. geiseri* (Fig. 1B) (*Hubbs, Edwards & Garrett 2008*). Both species have similar thermal tolerances (*Hagen, 1964*) and mostly abutted distributions in spring-river systems (*Watson, 2006*; *Behen, 2013*). If water temperature mediates feeding and therefore potential interactions, we predict *G. geiseri* will eat first and eat more food items than *G. affinis* at 23 °C and *G. affinis* will eat first and eat more food items than *G. geiseri* at 30 °C. Ability to identify stenothermic aquatic organisms and quantify temperature-mediated segregation will benefit our understanding of distributional patterns and improve threat assessments.

## MATERIALS AND METHODS

Laboratory specimens were collected with a seine from the Guadalupe River drainage basin under Texas Parks and Wildlife Scientific Research Permit No. SPR-0601-159. *G. geiseri* were collected from a site (29°53′22.2″N, 97°56′03.7″W) on the San Marcos River. *G. affinis* were collected from a site (29°54′43.8″N, 97°53′50.3″W) on the Blanco River approximately 13 river kilometers away from the *G. geiseri* collection site. Both species were collected from respective sites within the same day. Sexually-mature *Gambusia* >20 mm in total length (*Stevens, 1977*; *Haynes & Cashner, 1995*) were retained. Fishes were transported using insulated 52-L coolers to a laboratory at Texas State University Freeman Aquatic Biology Building within 30 min of capture. Within the laboratory, fishes were drip acclimated for 24 h to 23 °C with well water from the Edwards Aquifer, which is the same water source as the San Marcos and Blanco rivers (*Groeger et al., 1997*) and followed approved Texas State University Institutional Animal Care and Use Committee protocol (approval number: 201658034). Fishes were separated by species and placed into 35-L glass aquaria submersed in a LS-700 Living Stream (Frigid Units, Inc., Toledo, OH, USA). Each aquarium was equipped with a sponge filter. The Living Stream utilized a dual feedback heating and cooling system to maintain desired temperature within ±0.5 °C. Photoperiod was 14 h light:10 h dark. To maintain water quality, 50% water changes by volume were completed every 48 h. Fishes were fed high protein BioDiet Grower 1.2 mm (Bio-Oregon, Longview, WA, USA) daily ad libitum. To avoid any learned feeding behaviors, fishes were fed at varying times throughout the day and various locations of the aquaria. For 23 °C feeding trials, food was withheld 24 h prior to feeding trials. For 30 °C feeding trials, water temperature was adjusted 1 °C per day for 7 days (*Carmona-Catot, Magellan & Garcia-Berthou, 2013*). Fishes remained at 30 °C for 48 h before use in feeding trials, and food was withheld 24 h prior to feeding trials. Fishes were kept in the laboratory for a total of 3 days for the 23 °C trial, and 10 days for the 30 °C trial.

For pairwise feeding trials, one *G. geiseri* and one *G. affinis* were visually sized (i.e., within five mm) and sex matched (*Carmona-Catot, Magellan & Garcia-Berthou, 2013*), placed into a 1.25-L opaque container (23 × 15 cm area), and allowed to acclimate for 1 h. The container was immersed in the Living Stream to maintain the target temperature within ±0.5 °C. Five natural prey items (Order Trichoptera, Family Hydroptilidae; *Sokolov & Chvaliova, 1936*) were placed into the center of the container using a plastic pipette. The species of the individual feeding (i.e., strikes that consumed all, part, or none of the prey) first was recorded and total feeds were recorded for both individuals. Each pairwise trial was limited to 5 min or until all food items were consumed. After completion, fishes were euthanized in MS-222 (Tricane-S) and preserved in 10% formalin; therefore, a fish was used only once in a feeding trial. For a no feed trial, the two individuals were given an additional 30 min to acclimate and tried for an additional trial. Four independent test batches were conducted at 23 and 30 °C. A test batch was defined as all successful feeding trials at a certain temperature conducted within a 4–6 h period. All test batches were conducted between May 2017 and July 2017.
**Table 1 Number of first feeds and mean number of total feeds by batch for *Gambusia geiseri* and *Gambusia affinis* at 23 and 30 °C.**

| Temperature (°C) | Batch | N of pairs | Number of first feeds | | Mean number of total feeds | |
|---|---|---|---|---|---|---|
| | | | *G. geiseri* | *G. affinis* | *G. geiseri* | *G. affinis* |
| 23 | 1 | 7 | 5 | 2 | 2.29 | 0.29 |
| | 2 | 6 | 4 | 2 | 2.00 | 0.83 |
| | 3 | 6 | 5 | 1 | 1.60 | 0.33 |
| | 4 | 11 | 6 | 5 | 1.70 | 1.80 |
| | Mean | | 5.00 | 2.50 | 1.90 | 0.81 |
| | SD | | 0.82 | 1.73 | 0.31 | 0.70 |
| 30 | 1 | 9 | 3 | 6 | 1.44 | 2.11 |
| | 2 | 7 | 4 | 3 | 1.43 | 2.71 |
| | 3 | 7 | 0 | 7 | 0.00 | 4.29 |
| | 4 | 8 | 3 | 5 | 0.88 | 2.00 |
| | Mean | | 2.50 | 5.25 | 0.94 | 2.78 |
| | SD | | 1.73 | 1.71 | 0.68 | 1.05 |

Targeted number of pairwise matches was 10–11 per batch, but fish jumped out of the container on four occasions, and neither fish eating after 30 min occurred on 16 occasions. For both instances, the pairwise trial was ended and recorded observations were discarded.

Number of first feeds and mean number of total feeds were calculated for each species by target temperature. Number of first feeds was calculated by summing the number of first feeds by species per batch. Mean number of total feeds was calculated by summing of total feeds by species in each batch and dividing by the number of pairwise trials. One tailed two sample *t*-tests (SAS Institute, Gary, NC, USA) were used to detect differences in first feeds and mean number of total feeds between species at 23 and 30 °C. Use of one tailed *t*-tests were justified by the *a priori* prediction that the spring-associated fish would outperform at 23 °C and the riverine-associated fish would outperform at 30 °C.

## RESULTS

At 23 °C, 30 pairwise first feeds and 82 total feeds were observed out of 150 food items offered between *G. geiseri* and *G. affinis* pairs among four batches. Number of first feeds was greater ($t = 2.61$, d$f = 6$, $P = 0.02$) for *G. geiseri* than *G. affinis* (Table 1). Mean number of total feeds was greater ($t = 2.82$, d$f = 6$, $P = 0.02$) for *G. geiseri* than *G. affinis*.

At 30 °C, 31 pairwise first feeds and 111 total feeds were observed out of 160 food items offered between *G. geiseri* and *G. affinis* pairs among four batches. Number of first feeds was greater ($t = 2.26$, d$f = 6$, $P = 0.03$) for *G. affinis* than *G. geiseri*. Mean number of total feeds was greater ($t = 2.94$, d$f = 6$, $P = 0.01$) for *G. affinis* than *G. geiseri*.

## DISCUSSION

Our predictions that spring-associated *G. geiseri* has greater feeding performance than riverine-associated *G. affinis* at a temperature typical of a spring complex, and conversely, *G. affinis* has greater feeding performance than *G. geiseri* at a temperature typical of a

summertime riverine environment were supported by pairwise trials. These results are similar to temperature-mediated feeding performances among families, genera, and species reported by others (*De Staso & Rahel, 1994*; *Taniguchi et al., 1998*; *Taniguchi & Nakano, 2000*; *Carmona-Catot, Magellan & Garcia-Berthou, 2013*). This study, however, is novel in that it documents temperature-mediated performance between a spring-associated fish and a riverine-associated fish with similar thermal tolerances (*Hagen, 1964*). Greater feeding performance of *G. affinis* at a water temperature of 30 °C corresponds with the reported fastest growth rates and greatest natality rates of *G. affinis* at 30 °C when compared to 20 and 25 °C (*Vondracek, Wurtsbaugh & Cech, 1988*). Our results and the findings of *Vondracek, Wurtsbaugh & Cech (1988)* suggest that a warmer water temperature increases physiological and feeding performance of *G. affinis*, which corresponds with distributions of *G. affinis* during summertime in riverine environments. Conversely, a water temperature typical of spring complexes increases feeding performance for *G. geiseri*, which corresponds with distributions of *G. geiseri* during summertime in spring complexes. Growth rates, natality rates, and other measures of physiology are not known at this time for *G. geiseri*. Additionally, influences of other abiotic differences between spring systems and riverine systems (e.g., pH, specific conductance, and turbidity; *Groeger et al., 1997*) have not been assessed as to their role underlying *G. geiseri* and *G. affinis* segregation.

In order to show ubiquity of temperature-mediated performance as a mechanism for segregation among species distributions, feeding comparisons in addition to quantification of other temperature-mediated performance measures (e.g., growth and swimming performance) can be assessed for several other closely related taxa with similar distributions as *G. geiseri* and *G. affinis* within spring-river systems, such as spring-associated *Etheostoma lepidum* (*Hubbs, 1985*) and riverine-associated *E. spectabile*, *E. fonticola* (*Bonner & McDonald, 2005*) and *E. proeliare*, *Cyprinella proserpina* (*Hubbs, 1995*) and *C. lutrensis*, *Dionda argentosa* (*Garrett, Hubbs & Edwards, 2002*) and *D. diaboli*, and *Ictalurus lupus* (*Sublette, Hatch & Sublette, 1990*) and *I. punctatus*. In addition, spring-associated fishes and riverine-associated fishes maintain segregation during the winter when water temperatures of riverine environments are colder than water temperatures of spring complexes (*Kollaus & Bonner, 2012*). Assessments of feeding performance among spring-associated fishes and riverine-associated fishes at typical winter time temperatures would complete the range of conditions in which segregation is maintained. Ultimately, quantification of genetic, physiological, and biochemical mechanisms will be necessary to describe underlying temperature-mediated performance of spring-associated and riverine-associated species (see review in *Somero, Dahlhoff & Lin, 1996*). At a minimum, known mechanisms for stenotherm radiation and maintenance can serve as a basis for understanding evolutionary origins and maintenance of segregation among spring-associated and riverine-associated fishes.

Ability to identify stenothermic aquatic organisms and to quantify temperature-mediated segregation will benefit our understanding of distributional patterns and improve threat assessments. Stenothermic organisms are potentially more sensitive to temperature changes related to physical habitat alterations and global climate change than eurythermic

organisms because of the lack of gene product selection associated with eurythermic organisms (*Somero, Dahlhoff & Lin, 1996*). Physical habitat alterations include instream or riparian modifications that manipulate the energy budget or thermal capacity of the surface water (*Poole & Berman, 2001*), such as discharge of heated effluents (*Langford, 1990*; *Rahel & Olden, 2008*), removal of riparian vegetation (*Moore, Spittlehouse & Story, 2005*), stream channel modification (*Nelson & Palmer, 2007*), dams and diversions (*Olden & Naiman, 2010*), and reduction of discharge through groundwater pumping (*Sinokrot & Gulliver, 2000*). Groundwater sources supporting spring complexes of southwest USA are commodities (*Loáiciga, 2003*), and groundwater harvest is linked to the loss of spring complexes and associated biota (C.A. Craig & T.H. Bonner, 2018, unpublished data; *Winemiller & Anderson, 1997*). Within the Edwards Plateau, there are eight federally listed and 12 Texas state listed fishes that are associated with spring complexes. Our manuscript supports why spring-associated fishes are found within spring complexes, which informs natural resource managers in supervising species and their habitats (e.g., minimize groundwater withdrawals during periods of natural low flow to maintain stenothermal habitats for spring fishes). Continued climate change in North America is predicted to alter stream flow patterns, increase storm events, decrease dissolved oxygen, and increase groundwater temperatures (*Poff, Brinson & Day, 2002*; *Ficke, Myrick & Hansen, 2007*). As with physical habitat alterations, stenothermic aquatic organisms are predicted to follow isoclines of suitable habitat (*Ficke, Myrick & Hansen, 2007*), remain in place and wait for better times, adapt to changes, or become extinct (*Clarke, 1996*).

## CONCLUSIONS

This study supports a prediction that temperature mediates species distribution of a spring-associated and a riverine-associated fish through laboratory trials. Novel results of this study show temperature-mediated feeding performance of two species with similar temperature tolerances that inhabit spring-river systems. Although further work is needed to test for the ubiquity among other fishes, this study suggests temperature to be a structuring mechanism for organisms in spring-river systems.

## ACKNOWLEDGEMENTS

We would like to thank numerous graduate and undergraduate students for assistance with field collections. We would also like to thank C. Gabor for suggestions and guidance on concept and study design.

### Funding

The authors received no funding for this work.

### Competing Interests

The authors declare that they have no competing interests.

## Author Contributions

- Cody A. Craig conceived and designed the experiments, performed the experiments, analyzed the data, contributed reagents/materials/analysis tools, prepared figures and/or tables, authored or reviewed drafts of the paper, approved the final draft.
- Jeremy D. Maikoetter conceived and designed the experiments, performed the experiments, analyzed the data, contributed reagents/materials/analysis tools, prepared figures and/or tables, authored or reviewed drafts of the paper, approved the final draft.
- Timothy H. Bonner conceived and designed the experiments, analyzed the data, contributed reagents/materials/analysis tools, prepared figures and/or tables, authored or reviewed drafts of the paper, approved the final draft.

## Animal Ethics

The following information was supplied relating to ethical approvals (i.e., approving body and any reference numbers):

Full approval by the Texas State Universiy Institutional Animal Care and Use Committee: IACUC: 201658034.

## Field Study Permissions

The following information was supplied relating to field study approvals (i.e., approving body and any reference numbers):

Fishes were taken under Texas Parks and Wildlife Scientific Research Permit No. SPR-0601-159.

## Data Availability

Results from feeding batches are available in Table 1.

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
