# Peer review of "Temperature-mediated feeding between spring-associated and riverine-associated congeners, with implications for community segregation"

_PeerJ, doi:10.7717/peerj.6144_

## Round 0.1 · original submission · Major Revisions

Both reviewers think the study is interesting but not very compelling in scope. However, I am willing to look closely at a revision that effectively deals with the issues raised by both reviewers.

Reviewer 1 ·

Basic reporting

The authors use clear and unambiguous English but there are few minor errors throughout the text. For example: Line 23: “trails” should be “trials”. Line 39: delete the s from Temperatures. Line 81: “Purpose of this study…” should read “The purpose of this study…”

The introduction of the paper gives sufficient background to show the context of the work. It is clear how the authors are building on previous research.

The literature is well referenced and shows the relevance of the study. However, there are many errors in the text of the Literature Cited Section (see below). I recommend thoroughly checking all references.
Line 232. Behen: Reference incomplete.
Line 290. Poff, Brinson and Day: reference is not complete.
Line 295. Portner, Peck and Somero: Reference has a repeated information.
Line 306. De staso: reference is out of place and repeated here for a second time
Line 309. Stevens: Reference incomplete.
Line 324. Watson: Reference incomplete.

The manuscript structure conforms to PeerJ standards.

There are absolutely no figures in this paper, this is very strange. The authors should include: Fig. 1 - a map of the study location. This should include an inset of the relative location of each site. Fig. 2 – a picture panel of the two species or perhaps of the containers that they were placed in for observation. Fig. 3 - a basic graph of the feeding results.

All raw data on first and total feeding attempts has been supplied and is part of Table 1.

Experimental design

This paper contains original primary research on temperature-mediated species performance of two fish species, one spring and one river-associated. Dissimilar to previous research this manuscript focuses on two species that are both warm water forms with similar temperature tolerances, but that have different habitat preferences.

The research question is well defined. Past research has focused on temperature-mediated performance between warm and cold water forms (that change in distribution over altitudinal gradients). This study tested whether temperature-mediated performance might be a mechanism to segregate the habitat use of two warm water forms that have similar temperature tolerances. The authors state how their research fills an identified knowledge gap.

Additional details in the methods section are required to provide enough information for the study to be replicated. For example, were the different species collected from the different sites on the same days? How were the fish transported to the lab? How long in total were they in the lab before experiments were performed? How long did it take to drip acclimate them to 23 degrees C?

Validity of the findings

The feeding data are sufficient to address the question. However, data should be presented on the sizes of the fish to confirm that the visual matching of size and sex was sufficient to make the comparisons and that there was not a size bias by species.

The conclusion states that temperature mediates the distribution of these species. I think that it should be worded less strongly because other factors are also influencing the distribution of these species. Furthermore, this was just one feeding study at different temperatures and it didn’t include actual measurements of physiology, etc, so don’t conclude too much. The conclusions are linked to the original research question.

Additional comments

This manuscript is clear and concise. It flows logically, states a prediction, and presents the data set to support the prediction. A few things should be improved:
1. Your most important issue is the lack of figures in the manuscript.
2. Secondly the literature cited section needs to be revised to ensure completion of all references.
3. Please add some more details in the methods section.
4. Additional results on the sizes of the fish in the trials should be reported.

·

Basic reporting

This article is clearly presented and the context is sufficient. The structure is professional, the data are clearly presented and the hypotheses are clearly articulated.

Experimental design

The experimental design is sound in many ways. It is not very original. The only novel component is in the specific fish species that they used. The basic design and motivation mimic other studies. The overall research question is relevant and meaningful but the study addresses only a small component of the overarching goal. The research is technically sound and meets a high ethical standard. The methods would be easy to replicate.

Validity of the findings

The data are robust and statistically sound. The conclusions are supported with respect to the specific hypothesis but because the experiment is limited in scope, the conclusions in light of the over all theme are overstated. The data collected are very specific and do not adequately address the larger context articulated in the introduction.

Additional comments

This paper presents results from a laboratory study testing how temperature affects feeding performance in two small fish. The conceptual framework is well developed and the predictions and hypotheses are well articulated. The research itself is modest. The study is straightforward, testing if fish feeding performance differ for two species at different temperatures. The methods are sound all though they are not novel. The small data set is clearly presented in a Table with appropriate statistical analysis. The results support the hypothesis that the fish species collected from springs performs better at the temperature of springs and the fish species collected from the river fish performs better at the temperature of the river.


In addition to the small scope of this paper I have concerns with the interpretation of the results, given the experimental design. Because these are paired trials - there is an inherent assumption of competition yet there is no data on performance of the two species in isolation. This study would be stronger if it were coupled with individual feeding trials to test if there is an underlying physiological basis for the difference in performance of the two species with temperature. As the title suggests the authors are ultimately looking to understand community segregation - inferring that the underlying mechanism for separation is different competitive abilities under different habitat regimes.

The experiment is somewhat contrived as there is only one food source, food is limiting and species are forced to interact in a small space. The food was almost entirely consumed during some of the trials. It is unclear if the mean number of food items consumed is driven more by food limitation then by the rather forced constraints of feeding. The authors need to address how this study is relevant to field conditions.

On a more detailed note, the results are largely driven by one trial run where one of the species didn't feed at the high temperature. Given that these results are so different from the other trials it makes me wonder if there was something else going on with this batch of fish. Could they have been more stressed than the other fish?

This research would be stronger if the experiment were coupled with growth rate measurements, other behavior or physiology data to give a broader picture of fish performance under different temperature regimes. The authors develop predictions based on contrasting stenothermal vs. eurythermal environments but set up an experiment to test two temperatures. Testing how fluctuations in temperature affect fish distributions requires a more elaborate design.

In summary - the results from this study are compelling in that they support the predictions articulated by the authors. The scope of the study however is limited. This research should either be condensed into a short note or submitted to a regional journal.

---

## Round 0.2 · Minor Revisions

The paper is very close to acceptance.

I agree that Figure 2 is not needed (redundant to the Table), so leave it out of your final revision. I think the photo of the fish is helpful so leave it in.

I want you to consider integrating a couple of older papers into your introduction. They deal with pupfish in thermal spring streams in Death Valley if I remember correctly, pretty similar to your study I think and should be cited (ok to push back on that if you don't agree of course). The key papers are:

Productivity of a herbivorous pupfish population (Cyprinodon nevadensis) in a warm desert stream
RJ Naiman - Journal of Fish Biology, 1976

Effects of constant and fluctuating temperatures on reproductive performance of a desert pupfish, Cyprinodon n. nevadensis
JB Shrode, SD Gerking - Physiological Zoology, 1977

Finally, consider beefing up the relevance statements in your discussion along the lines of the statements in your rebuttal letter -e.g., importance to Edwards Plateau.

I apologize for the delay on these very minor comments. We try to be prompt at PeerJ but I was traveling.

---

## Round 0.3 · accepted · Accept

The response to reviews improved the paper.

#